# Review of Novel Surgical, Radiation, and Systemic Therapies and Clinical Trials in Glioblastoma

**DOI:** 10.3390/ijms251910570

**Published:** 2024-09-30

**Authors:** Allison R. Valerius, Lauren M. Webb, Anna Thomsen, Eric J. Lehrer, William G. Breen, Jian L. Campian, Cecile Riviere-Cazaux, Terry C. Burns, Ugur Sener

**Affiliations:** 1Department of Neurology, Mayo Clinic, Rochester, MN 55905, USAsener.ugur@mayo.edu (U.S.); 2Department of Radiation Oncology, Mayo Clinic, Rochester, MN 55905, USA; 3Department of Medical Oncology, Mayo Clinic, Rochester, MN 55905, USA; 4Department of Neurosurgery, Mayo Clinic, Rochester, MN 55905, USA

**Keywords:** glioblastoma, high-grade glioma, immunotherapy, targeted molecular therapy, surgical therapy, radiation therapy, review of therapy

## Abstract

Glioblastoma (GBM) is the most common malignant primary brain tumor in adults. Despite an established standard of care including surgical resection, radiation therapy, and chemotherapy, GBM unfortunately is associated with a dismal prognosis. Therefore, researchers are extensively evaluating avenues to expand GBM therapy and improve outcomes in patients with GBM. In this review, we provide a broad overview of novel GBM therapies that have recently completed or are actively undergoing study in clinical trials. These therapies expand across medical, surgical, and radiation clinical trials. We additionally review methods for improving clinical trial design in GBM.

## 1. Introduction

Glioblastoma (GBM) is the most common malignant primary brain tumor in adults, accounting for 14.2% of all brain tumors and 50.9% of all malignant brain tumors [1]. Standard treatment for newly diagnosed GBM includes maximal safe surgical resection followed by six weeks of radiation therapy (RT) with concurrent and adjuvant temozolomide (TMZ) for six total cycles. Hypofractionated radiation therapy may be used depending on factors such as patient age or performance status. Additional FDA approved therapies for newly diagnosed GBM include tumor treating fields (TTFs). TTFs may not be available in all clinical settings and are limited to patient tolerability, but have received FDA approval after a significant improvement in overall survival (OS) (20.9 months versus 16.0 months; *p* < 0.001) [2]. Greater extent of resection (EOR) achieved at initial surgery and epigenetic silencing of DNA-repair gene O6-methylguanine–DNA methyltransferase (MGMT) are factors associated with improved survival [3]. Despite multimodality treatment for newly diagnosed GBM, recurrence is universal and GBM prognosis is poor with a five-year survival rate of 6.9% among adults [1].

To date, no single treatment has been demonstrated to confer a survival benefit for recurrent GBM in a large-scale clinical trial that has resulted in a widespread change in clinical practice [4,5,6,7,8]. Therefore, there is currently no standardized treatment for recurrent GBM, and, if available, enrollment in a clinical trial is highly encouraged [9]. If clinical trials are unavailable, National Comprehensive Cancer Network (NCCN) guideline-based treatments that can be considered for recurrent GBM include bevacizumab, lomustine, temozolomide rechallenge, or a combination of bevacizumab with TMZ or lomustine [9,10,11]. Carmustine wafers can be considered but are rarely utilized in clinical practice due to unclear survival benefit and risk of wound complications [12]. Repeat surgery or additional RT can be considered, but these interventions have not been associated with a clear survival benefit [13,14]. In this review, we discuss innovations in surgical management, radiation therapy, systemic therapy, and trial design that may shift this paradigm and ultimately improve survival rate in newly diagnosed and/or recurrent GBM.

## 2. Innovations in Surgical Management

Maximal safe surgical resection is the starting point for GBM treatment [15]. Traditionally, the goal of surgery has been to achieve maximal resection of the enhancing tumor, and it is known that gross total resection (GTR) of enhancing disease confers a survival benefit [16,17]. More recent studies suggest potential value of resecting beyond the T1-contrast enhanced (T1CE) tumor on MRI in high-grade gliomas [18,19]. In addition, a study of 1047 patients suggested residual tumor volume (RTV) rather than EOR may be more predictive of overall survival (OS) and progression-free survival (PFS) [17]. The longest survival times were observed in patients with both a radiographic GTR and preservation of neurological function [20]. These findings underscore the importance of novel techniques that can maximize EOR and minimize RTV while maintaining neurological function [20]. As surgical intervention is typically the first GBM-targeting treatment to occur, improvements in the extent and/or success of surgical intervention, as discussed in the following sections, are an important contributor to improving patient outcomes. Novel surgical interventions are reviewed in Table 1.

### 2.1. Neuronavigation, Intraoperative MRI, and Intraoperative Ultrasound

Neuronavigation, such as Stealth™ or BrainLab™, uses pre-operative magnetic resonance imaging (MRI) obtained with fiducials or facial tracing to define the location of anatomical structures prior to and during surgery. Intraoperative MRI (iMRI) can provide updated imaging during surgery after brain shift has occurred, ensuring optimal resection prior to closure, or maximal accuracy if needed to resect the final portions of remaining disease near critical structures [21]. In a prior meta-analysis, iMRI was superior to conventional neuronavigation alone for achieving radiographic GTR of GBM [22]. Intraoperative ultrasound can also provide real-time anatomical updates, but its ability to discriminate contrast enhancing from non-enhancing disease is limited [21].

### 2.2. Fluorescence-Guided Surgery

As GBMs are diffusely infiltrative tumors, differentiation of non-enhancing tumor from vasogenic edema is challenging. Fluorescence-guided surgery, including 5-aminolevulinic acid (5-ALA) or sodium fluorescein, can significantly improve EOR in GBM [23]. 5-ALA accumulates in glioma cells after oral administration and is metabolized into protoporphyrin IX (PpIX), which is visible under fluorescent light at 630 nm [24]. In a randomized phase III study, use of 5-ALA improved EOR as compared to conventional microsurgery [23]. 5-ALA is well-tolerated, with only mild, reversible adverse effects including photosensitivity requiring minimal light exposure for 48 h after administration [24,25]. Sodium fluorescein has also been evaluated as a lower-cost intravenous fluorescent agent to guide resection in GBM. Sodium fluorescein is a BBB-impenetrant compound and therefore highlights only contrast-enhancing disease [26]. To date, no large-scale trials have compared 5-ALA to fluorescein. Other fluorescent agents, such as indocyanine green or endogenous fluorophores, are under investigation [27].

### 2.3. Intraoperative Mapping

GBM often infiltrates into eloquent brain areas critical for speech, motor function, and cognitive performance, making surgery that minimizes post-operative neurologic deficits challenging. Awake surgery can be performed while directly monitoring motor, sensory, language, or cognitive function [28,29]. Function can be located by electrical stimulation of cortex or subcortical white matter tracts to assess the safety of resecting a specific region. Ultrasonic aspiration can inactivate adjacent brain tissue providing advance warning prior to reaching eloquent structures [30]. Proximity to corticospinal tracts can be evaluated via direct motor stimulation eliciting EMG activity in asleep patients [31]. Any combination of these approaches can be utilized to balance the risks of post-operative neurological deficits with the benefit of achieving minimal RTV [32]. Risks of intraoperative mapping include loss of airway or intraoperative seizures [29]. As mapping tools grow more advanced, and our understanding of brain functional circuitry and neuroplasticity increases, intraoperative mapping is likely to become more nuanced.

### 2.4. Confocal Microscopy, Raman, and Mass Spectroscopy

Intraoperative pathology is often limited to a nonspecific diagnosis of glioma, requiring further post-operative histological and genetic evaluation. Real-time evaluation for the presence of glioma cells may help guide the EOR at tumor margins. Confocal laser technology can provide high-resolution imaging of glioma tissues to identify vascular neoproliferation and tumor margins [33]. In vivo confocal microscopy correlates positively with traditional histological findings [33]. Raman spectroscopy is a technique that measures light scattering to generate a molecular fingerprint for a sample. When combined with deep neural network training, intraoperative stimulated Raman histology was non-inferior to a pathologist-based evaluation of histology (94.6% vs. 93.9% accuracy) and could molecularly classify gliomas based on IDH-mutation, 1p19q co-deletion, and ATRX mutations with an accuracy of 93.3% [34]. AI-based methods may also provide highly accurate estimates of molecular disease classifications based on H&E sections with intraoperative utility [35]. Further work is needed to prospectively evaluate the utility of these techniques.

### 2.5. Phase 0/Window-of-Opportunities Studies

Clinical trials in patients with GBM have been limited by minimal tissue feedback regarding on-target therapeutic impact and intratumoral drug concentrations. To obtain critical data on whether a therapy is reaching the tumor, patients can receive a drug or drug combination prior to a clinically indicated biopsy or resection via a phase 0 or window-of-opportunity study. Such studies have been utilized to evaluate intratumoral drug concentrations of novel agents such as tyrosine kinase inhibitors and immunotherapies [36]. Post-drug tissue can be assayed to determine the agent’s pharmacodynamic impact on the tumor [36]. These early phase studies can provide early readouts on the potential effectiveness of a candidate therapy. However, GBMs are highly heterogeneous tumors, which can hinder evaluation of pharmacodynamic impact [36]. Baseline tissue samples via biopsies prior to drug administration and subsequent resection can facilitate analysis of drug-induced impacts. For newly diagnosed disease, baseline tissue may help guide patients to neoadjuvant trials of the most relevant therapy. For recurrent disease, tissue from biopsy can help adjudicate imaging findings equivocal for true disease progression versus treatment-induced radiographic changes [37]. The resulting paired pre- and post-treatment tissue samples must be thoughtfully collected, ideally from multiple locations, to circumvent confounders of intra-tumoral heterogeneity to provide powerful insights to accelerate therapeutic development. Serial biopsies obtained longitudinally throughout therapy provide opportunities for unprecedented biological feedback [38].

### 2.6. CSF Access for Longitudinal Monitoring

While tissue is only accessible at the time of biopsy or resection, cerebrospinal fluid (CSF) can be obtained longitudinally via lumbar punctures and/or intracranial CSF access devices, such as Ommaya reservoirs, that can be placed at the time of biopsy or resection. The ability to longitudinally evaluate candidate CSF biomarkers, including 2-hydroxyglutarate, cell-free DNA, and proteins has been demonstrated [39,40,41]. Further research is needed to evaluate the sensitivity and specificity of CSF-based approaches for disease monitoring in larger patient cohorts. Serial CSF sampling may also inform pharmacodynamic impacts of candidate therapies during clinical trials. For example, a recent study demonstrated significant changes in cytokines and chemokines following intrathecal bivalent chimeric antigen receptor (CAR) T-cells targeting EGFR and IL13R α2 [42].

### 2.7. Local Drug Delivery

Systemic toxicity and limited drug distribution across the blood brain barrier (BBB) may prevent agents from reaching intratumoral therapeutic concentrations. To bypass the BBB, chemotherapies can be intrathecally delivered via the lumbar cistern or intraventricular/intracavitary delivery using Ommaya reservoirs. However, drug delivery directly in the CSF may still be limited due to the blood–ependymal barrier, non-targeted distribution, and poor tissue penetration [43]. Local injection of experimental therapies, such as CAR T-cells, or oncolytic viruses, has been performed in patients with GBM [44].

Intracavitary wafers, such as Gliadel^®^ wafers delivering carmustine, have been utilized to locally deliver drugs around the resection cavity. Gliadel^®^ wafers are FDA-approved for both newly diagnosed GBM and recurrent GBM as an adjunct to resection after a clinical trial of 248 newly diagnosed patients demonstrated a modest survival benefit of 13.9 versus 11.6 months (*p* = 0.03) [45]. However, limited drug diffusion, difficulty in keeping wafers stable within the resection cavity, and wound infection concerns have limited the widespread use of this innovation.

Convection enhanced delivery (CED) is another technique to improve drug distribution throughout a GBM. CED is a bulk flow-driven process wherein catheters are placed in a region of interest and perfused with a drug under positive pressure to bypass diffusion processes [43]. Prior trials have demonstrated the safety and feasibility of CED but have not shown significant improvement in survival [43]. Technical challenges can include drug reflux around the catheter at high positive pressures, as well as homogeneous coverage of heterogenous tumors with intrinsically elevated interstitial fluid pressure [46]. Additionally, direct drug delivery to the CNS does not preclude efflux through transporters, underscoring the importance of thoughtful drug selection for CED [47].

### 2.8. Focused Ultrasound

Methods that increase BBB permeability may augment intratumoral drug concentrations. Focused ultrasound (FUS) achieves this through application of ultrasound energy in combination with intravenous microbubbles to temporarily disrupt the BBB [48,49]. Clinical trials are currently evaluating the ability of FUS to increase CNS penetration of agents such as carboplatin and albumin-bound paclitaxel [50]. Two studies are evaluating sonodynamic therapy, which utilizes a sono-sensitizing agent (e.g., 5-ALA, fluorescein), that creates reactive oxygen species via PpIX excitation to kill tumor cells [51]. One study is currently evaluating FUS therapy as a radiosensitization strategy based on generation of highly accurate transcranial hyperthermia [52]. Since the BBB can also limit diffusion of CNS-derived biomarkers into plasma, FUS may be of use to improve the release of glioma-associated biomarkers into the systemic circulation for disease monitoring. The applicability of MRI-guided FUS may be limited in part by how frequently it can be performed due to the length of the procedures [53]. Implantable FUS, such as those by Carthera^®^, may enable more frequent BBB disruption without need for MRI guidance and is currently under evaluation [54]. However, implantable FUS arrays cannot be adjusted after implantation to treat tumor that extends beyond the sonication target volume [50].

### 2.9. Laser Interstitial Thermal Therapy (LITT)

LITT is a minimally invasive ablative technique that uses heat to damage cancer cells. Although its applicability in GBM is currently limited due to its small treatment volume, advances in LITT to enable larger treatment volumes may provide an avenue for LITT in glioma management. LITT may also provide a transient window of enhanced BBB permeability and is under investigation as an adjunct to immunotherapy [55].

## 3. Innovations in Radiation Therapy

Conventional radiation therapy (RT) kills tumor cells by utilizing photons from X-rays to create nonspecific breaks in the DNA strands of rapidly dividing cells [56]. Standard RT for GBM involves administration of 60 Gy in 30 fractions over 6 weeks with concurrent and adjuvant TMZ. Various alterations to the standard of care photon RT, such as enhanced tumor visualization for radiation planning, are under investigation, as are novel radiation therapies (Table 2).

### 3.1. Particle Based Therapy

#### 3.1.1. Proton Radiotherapy

Photons deposit energy along the X-ray beam path, with the highest deposition near the point of entry [57]. Protons, in contrast, deposit increased energy as the particles slow and interact with surrounding tissue, depositing maximum energy at a depth specific to the tumor, referred to as the Bragg peak [58]. The relative biological effectiveness (RBE) of protons is slightly higher than that of photons [59]. Proton therapy can limit off-target doses and prevent side effects related to normal tissue toxicity [57]. The utility of proton therapy in GBM remains unclear, with varying data regarding survival benefit [60,61].

The use of advanced imaging to better target radiation, especially to non-enhancing disease, is an area of active study. Utilizing diffusion-weighted MRI and dynamic contrast-enhanced perfusion MRI to target hypercellular and hyperperfused tumor volumes, OS at 12 months among boosted patients was 92% [62]. F-DOPA PET imaging has been used to guide dose-escalated radiation therapy (DERT) [63]. In a cohort of 75 patients, F-DOPA PET-guided DERT appeared to be safe and requires further study to determine if OS or PFS are statistically significant [63].
ijms-25-10570-t002_Table 2Table 2Novel radiation therapy clinical trials in glioblastoma.Radiotherapy Trial IdentifierTrial TitleTumor TypeTrial PhaseTrial StatusTrial DetailsPhoton Radiotherapy NCT05781321Short Course Radiotherapy for the Treatment of Patients with Glioblastoma, SAGA StudyGBMIIActive, RecruitingRandom allocation to 1 of 2 arms:
1.Arm A: short course RT (5–10 fractions over 1–2 weeks) of higher doses of radiation with concurrent and adjuvant TMZ.2.Arm B: standard course RT for 15–30 fractions over 3-weeks with concurrent and adjuvant TMZ.
Primary objective: 12-month overall survival (OS).Photon Radiotherapy—Dose-EscalatedMC1374NCI-2013-0224218F-DOPA-PET in Finding Tumors in Patients with Newly Diagnosed Gliomas Undergoing Radiation TherapyGrade IV Malignant GliomaIIPrimary Completed75 patients underwent ^18^F-DOPA PET imaging to guide dose-escalated radiation therapy (DERT). Primary objective was 6-month progression-free survival (PFS) compared to historical controls. Initial results indicated significant improvement in PFS in MGMT unmethylated GBM and significantly improved OS in MGMT methylated GBM patients [63].Proton RadiotherapyNCT02163135Dose-Escalated IMRT or Proton Beam Radiation Therapy Versus Standard-Dose Radiation Therapy and Temozolomide in Treating Patients with Newly Diagnosed GlioblastomaNew GBMIIActive, Not Yet RecruitingRandom allocation to 1 of 4 arms:
1.Arm A1 Control: standard-dose photon irradiation.2.Arm B: Dose-escalated photon IMRT.3.Arm A2 Control: standard-dose photon irradiation.4.Arm C: dose-escalated proton beam therapy.
Primary objective: OS.Proton Radiotherapy & Carbon Ion RadiotherapyNCT04536649Proton and Heavy Ion Beam Radiation vs. Photon Beam Radiation for Newly Diagnosed GlioblastomaNew GBMPhase IIIActive, Not Yet RecruitingRandom allocation with 1:1:1 to three groups:
1.Control Group—Standard Photon Radiotherapy.2.Study Group A—standard-dose proton radiotherapy.3.Study Group B—standard dose proton radiotherapy plus induction carbon-ion radiotherapy boost.
Primary objective: OS.Surgically Targeted Radiotherapy—GammaTileNCT04427384Registry of Patients with Brain Tumors Treated with STaRT (GammaTiles)All CNS tumorsIVActive, RecruitingObservational study. All patients will undergo maximum safe resection of intracranial neoplasm(s) and implantation of GammaTiles.Primary objectives: surgical bed-recurrence free survival, OS.GBM: glioblastoma; TMZ: temozolomide; IMRT: intensity-modulated radiation therapy.

#### 3.1.2. Carbon Ion Radiotherapy

Carbon ion radiotherapy (CIRT) is a heavy particle technique to elicit energy and induce double strand breaks in DNA [64]. CIRT has been proposed as an emerging therapy for radioresistant tumors, including GBM, as it has a higher RBE (1.1 to 3.74) compared to photons and protons [65]. Carbon particles can be configured to deliver maximal energy at specific tissue depths, minimizing off-target toxicities [64]. Carbon ions also induce more severe double strand breaks than photons and result in less effective tumor DNA damage repair, shown to be effective in glioma cell lines resistant to photons [64,66].

Multiple studies have suggested efficacy and tolerability of CIRT in patients with intracranial malignancies [67,68,69]. Ongoing studies include the CLEOPATRA trial (NCT01165671), a phase II study evaluating carbon ion boosts vs. photon boosts in patients with GBM [70]. CIRT is also under investigation in patients with recurrent GBM through the CINDERELLA trial (NCT01166308), a phase I/II study comparing CIRT to fractionated RT [71]. A multicenter phase III clinical trial for patients with newly diagnosed GBM (nGBM) comparing photon RT to proton RT with or without CIRT boost is ongoing (Table 2).

Current pitfalls of CIRT include its slow speed of treatment and subsequent theoretical margin of error due to patient movement, lack of long-term outcome data, and limited availability [64].

#### 3.1.3. Boron Neutron Capture

Boron neutron capture therapy (BNCT) is a new particle therapy utilizing a reaction between neutrons and a boron isotope (10B) [72]. After the boron compound is selectively taken up by malignant tumor cells, irradiation with neutrons kills the malignant boron-containing tumor cells without damaging the healthy, non-boron containing cells [72]. In a study by Yamamoto et al., 15 patients with newly diagnosed GBM received either intraoperative BNCT or external beam BNCT. The median OS was 25.7 months and the median time to tumor progression (TTP) was 11.9 months for all patients, demonstrating potential utility of this RT technique [73].

### 3.2. Surgically Targeted Radiation Therapies

#### 3.2.1. Intraoperative Radiotherapy (IORT)

Starting radiation at the time of initial surgery may improve local tumor control [44]. Various modalities of IORT have been studied, including intraoperative electron radiotherapy (IOERT) and low-energy X-ray intraoperative radiotherapy (LEX-IORT) but PFS or OS benefit is yet to be demonstrated [74,75].

#### 3.2.2. Interstitial Brachytherapy (IBT)

Interstitial brachytherapy includes use of small devices for delivery of RT to the resection cavity with radiation delivered over the weeks following placement [74]. Two historical randomized IBT trials conducted prior to modern classification of gliomas using iodine-125 for high-grade glioma patients did not demonstrate OS benefit [76,77].

A newer form of brachytherapy, GammaTile (GT), is a device consisting of cesium-131 radiation-emitting seeds in a collagen tile [78]. This surgically targeted radiation therapy exceeds the standard dose of external beam radiation therapy [79,80]. Implantable GT was FDA approved in 2018 for use in recurrent intracranial neoplasms and expanded to newly diagnosed intracranial neoplasms in 2020 based on a study of 20 high-grade previously irradiated meningiomas where PFS rate was 89% at 18 months from surgery with GT compared to 50% with surgery alone [79,81]. In a study of 40 patients with recurrent malignant gliomas, median local control duration after GT was 12 months [80]. Clinical trials are ongoing to better define the relative efficacy and safety of IBT compared to standard of care RT in patients with GBM (Table 2).

#### 3.2.3. Stereotactic Radiation Surgery (SRS)

SRS is the delivery of a highly conformal ablative dose of radiation to targets [82]. SRS often involves photon-based sources via dedicated radiosurgery platforms (Gamma Knife^®^, CyberKnife^®^) [82]. Post-operative SRS was evaluated in the RTOG 9305 trial with no OS benefit demonstrated. [83]. However, pre-operative SRS may have a role in GBM treatment. Pre-operative therapy would allow for more precise target delineation with potentially increased efficacy [84,85]. Pre-operative SRS for GBM is the subject of the ongoing NeoGlioma Study (NCT05030298).

The concurrent use of SRS and immunomodulators is also under investigation. The ablative doses used in SRS may allow for enhanced CD8+ T-cell activation with resultant antitumor immune response augmented by immune checkpoint inhibitors [86,87].

To reduce patient burden, researchers have started evaluating a 5-day course of SRS with 5 mm margins in nGBM patients with concurrent TMZ as an alternate to standard RT [88]. The initial phase I/II study found adverse radiation effects were limited to grade 1 or 2 and did not statistically impact survival [88].

### 3.3. Re-Irradiation

RTOG125 was a prospective, phase II, randomized trial of re-irradiation and bevacizumab (BEV) vs. BEV alone in patients with recurrent GBM (rGBM). Re-irradiation with bevacizumab conferred a PFS benefit compared to bevacizumab alone (7.1 vs. 3.8 months) but no OS benefit [89].

## 4. Innovations in Systemic Therapy

To date, TMZ remains the only systemic therapy associated with OS benefit in a large-scale clinical trial for GBM. Given the limited benefit of TMZ and no large-scale survival benefit associated with traditional salvage therapies, there is an urgent need for novel systemic therapeutics. Novel clinical trials in systemic therapy are summarized in Table 3.

### 4.1. Targeted Molecular Therapy

The Cancer Genome Atlas Network (TCGA) studies and other molecular profiling efforts provided tremendous insights into the intertumoral heterogeneity of GBM and identified potential mutations. Rather than exerting a direct cytotoxic effect as is the case with conventional chemotherapies, molecularly targeted therapies attempt to block pathways relevant to tumor growth.

#### 4.1.1. BRAF

V-RAF murine viral oncogene homolog B1 (BRAF) is a member of the Raf family of protein kinases that promotes cellular proliferation. BRAF inhibitors (BRAFi) are widely utilized in targeting the BRAF V600E mutation in non-CNS cancers [90,91]. Several types of BRAF mutations were also identified in GBM [92].

In a basket trial (NCT01524978), the BRAFi vemurafenib was associated with partial response (PR) in 1/11 and stable disease (SD) in 5/11 treated patients with *BRAF*-mutated glioma [93].

In the Rare Oncology Agnostic Research (ROAR) study, BRAFi dabrafenib with or without trametinib (a MEK inhibitor, MEKi) was evaluated in 45 patients with BRAF-mutated high-grade glioma (HGG, 31 diagnosed as GBM) [94]. Fifteen patients with HGG had an objective response (OR), including three complete responses (CRs) and 12 PRs [94].

In a systematic review and meta-analysis of BRAFi/MEKi in glioma,154 pediatric patients and 137 adult patients were included with CR/PR reported for 56% of pediatric and 38% of adult patients [95]. In June 2022, dabrafenib/trametinib combination therapy was approved by the United States Food and Drug Administration (FDA) for all solid tumors harboring BRAF V600E mutations, representing a treatment option for the small percentage of GBMs with this alteration [96].

#### 4.1.2. Neurotrophic Tyrosine Kinase Inhibitor (NTRK)

NTRK fusions have been identified in several types of cancer including 0.56–1.69% of GBMs [97]. Larotrectinib, a pan-TRK inhibitor, is approved by the FDA for adult and pediatric solid tumors with NTRK fusions [98]. Larotrectinib was studied in nine patients (two GBM) with primary CNS tumors with one achieving PR and seven achieving SD [99]. Larotrectinib is currently under evaluation for patients with primary CNS tumors (NCT02465060, NCT04142437).

Entrectinib, an NTRK, ALK, and ROS1 fusion inhibitor, is also FDA approved for solid tumors that have an NTRK gene fusion. In a combined analysis, CNS activity of entrectinib was suggested based on responses in one patient with glioneuronal tumor and one patient with lung cancer-related brain metastasis [100]. Entrectinib is being evaluated in adults with solid tumors, including GBM (NCT02568267).

#### 4.1.3. Epidermal Growth Factor Receptor (EGFR)

EGFR is overexpressed in approximately 60% of GBM and/or amplified in more than 40% of tumors with EGFRvIII representing the most common EGFR mutation in GBM, making it an appealing therapeutic target [101,102].

Depatuxizumab mafodotin, an EGFR antibody-drug conjugate was evaluated in nGBM with concurrent TMZ and demonstrated no survival benefit (NCT02573324) [103,104,105]. To date, EGFR targeting tyrosine kinase inhibitors (TKIs), such as erlotinib, have demonstrated no OS benefit in rGBM as a monotherapy [106]. Erlotinib combined with mammalian target of rapamycin (mTOR) inhibitor sirolimus demonstrated PR in 19% of patients in one study [102].

GC118, an EGFR receptor antibody was studied and demonstrated no PFS/OS benefit (NCT03618667) [107,108]. Bispecific T-cell engager AMG596 was evaluated in eight patients with GBM with one PR and two SD (NCT03296696) [108]. A newer small molecule EGFR inhibitor, ERAS-801, is undergoing evaluation for rGBM (NCT05222802) [109]. WSD0922-Fu, another EGFR inhibitor, is undergoing study in CNS tumors (NCT04197934) [110].

#### 4.1.4. PI3K-mTOR

There is loss of the tumor suppressor phosphatase and tensin homolog on chromosome ten (*PTEN*) gene in up to 60% of patients with GBM [111]. *PTEN* loss results in activating mutations of the PIK3CA and PI3K/mTOR molecular pathways, resulting in uncontrolled cell proliferation [101]. As a single agent, mTOR inhibitor temsirolimus was not associated with survival benefit in GBM [112]. Another mTOR inhibitor, everolimus, did not increase OS when combined with standard chemoradiotherapy [113].

Buparlisib, a pan-PI3K inhibitor, was studied in rGBM but was ineffective [101]. Paxalisib, a brain-penetrant small molecule PI3K and mTOR inhibitor has also been studied in GBM with initially promising results but was ineffective as part of a larger GBM trial [114,115,116].

#### 4.1.5. Vascular Endothelial Growth Factor (VEGF)

VEGF is a major mediator of angiogenesis in GBM [101]. Bevacizumab, an anti-VEGF antibody, has been extensively studied in GBM. Based on its PFS benefit and radiographic responses in 71% of patients in a prior study, bevacizumab was granted FDA approval for rGBM [117,118]. Unfortunately, subsequent randomized multicenter studies in nGBM and rGBM demonstrated no OS benefit from use of this antiangiogenic [119,120]. Nevertheless, bevacizumab is frequently used in clinical practice to manage vasogenic edema as a steroid-sparing agent and for its PFS benefit [121].

A different VEGF inhibitor, cediranib, also did not confer an OS benefit in randomized clinical trials [122]. Regorafenib, a VEGF inhibitor with affinity for multiple VEGF receptors and PDGF receptors, did significantly improve PFS and OS in patients with rGBM compared to lomustine in an initial study (NTC02926222), but subsequent study as part of the GBM Adaptive Global Initiative Learning Environment (AGILE) trial did not demonstrate OS benefit [123,124].

### 4.2. Immunotherapy

Immunotherapy is a rapidly expanding field in various malignancies, including GBM. Immunotherapy is a unique treatment approach as it aims to engage the patient’s own immune system to mount a sustained antitumor immune response rather than conventional RT and chemotherapy that seeks to directly kill tumor cells.

#### 4.2.1. Immune Checkpoint Inhibitors (ICI)

GBM is considered an “immunologically cold” tumor with a sparsity of infiltrating lymphocytes, an exhausted phenotype of tumor-infiltrating T-lymphocytes, and low tumor mutational burden to alert the immune system [125]. Additionally, GBM inhibits the immune system through immunosuppressive paracrine mediators and causes systemic immune system dysfunction [126,127].

ICIs have been studied to combat GBM’s immunosuppressive effects to generate an antitumor immune response. ICIs interfere with cell cycle checkpoint regulators like cytotoxic T-lymphocyte-associated protein 4 (CTLA-4) and programed cell death protein 1 (PD-1), which normally downregulate T-lymphocyte activation and promote self-tolerance to avoid autoimmunity [125]. However, to date, clinical trials with ICIs have not shown OS benefit for nGBM or rGBM. CheckMate 143 compared the anti-PD1 ICI nivolumab to bevacizumab for rGBM, demonstrating no OS benefit (NCT02017717) [128].

The addition of nivolumab to RT and TMZ was not associated with improved survival in MGMT-methylated or unmethylated nGBM (CheckMate 548 (NCT02667587) and CheckMate 498 (NCT02617589), respectively) [129]. Another anti-PD-1 ICI, pembrolizumab, was assessed with or without bevacizumab for rGBM in NCT02337491 [130]. Neither combined treatment nor pembrolizumab was effective with 6-month PFS rates of 26% vs. 6.7% and median OS 8.8 vs. 10.3 months [130].

Negative studies have led to the consideration of alternative strategies to utilize ICIs for GBM. In a small study of 35 patients with rGBM eligible for repeat resection, patients were randomized to neoadjuvant and adjuvant vs. adjuvant only pembrolizumab. Patients who received neoadjuvant ICI had longer OS compared to the adjuvant only group (417 vs. 229 days), but small sample size limits generalizability [131]. Higher tumor mutational burden may predict increased likelihood of response to ICI, which is the rationale for ongoing studies with pembrolizumab (NCT02658279) and combination therapy with ipilimumab plus nivolumab (NCT04145115) in recurrent gliomas with elevated mutational burden [114,115,125]. Because of higher PD-1 expression, patients with IDH wild-type glioma may be more responsive to ICIs than patients with IDH-mutant gliomas [127]. Other strategies under investigation include combination of ICI with RT, LITT, and other immunotherapies as well as evaluation of checkpoint inhibitors besides anti-PD-1 and anti-CTLA-4 targeting drugs [125,132,133].

#### 4.2.2. Vaccine-Based Therapies

Vaccine-based therapies are intended to invoke an anti-tumor immune response [125,134]. The heterogeneity of GBM and the evolution of tumor antigens overtime, termed “antigen escape”, present challenges for vaccine development [135].

#### Peptide Vaccines

Given frequent EGFRvIII mutations encountered in GBM, EGFRvIII-targeted peptide vaccine rindopepimut was developed and studied in a phase 3 trial [136]. Addition of rindopepimut to standard therapy for nGBM conferred no survival benefit the study was terminated for futility [137]. In subsequent analysis at progression, loss of EGFRvIII expression was noted in 57–59% of tumors in vaccine and placebo groups, illustrating antigen escape [125,126,138].

Survivin is an intracellular antiapoptotic peptide highly expressed in GBM cells but not in normal glial tissue [125,134]. Following promising results from a small study of nine patients with rGBM, SurVaxM, a peptide mimic vaccine that targets survivin, is under evaluation for rGBM in an ongoing study in combination with pembrolizumab (NCT04013672) [139].

Other peptide vaccines under investigation for GBM include Wilm’s tumor 1 (WT1) targeting DSP-7888 and VEGF receptor targeting VXM01 [140,141]. Multipeptide vaccines under investigation include IMA950/polyICLC, which contains nine peptide antigens from GBM samples and EO2401, which contains bacterial peptides mimicking antigens highly expressed in GBM [142,143].

#### Dendritic Cell Vaccines

Dendritic cells are the principal antigen-presenting cells (APC) responsible for enabling the transition from innate to adaptive immunity. Vaccines using dendritic cells have been studied for treatment of GBM [135]. The process involves isolating dendritic cells from the patient, exposing the dendritic cells ex vivo to target tumor antigens and cytokines for maturation, and then injecting the dendritic cells back into the patient, where APCs migrate to lymph nodes and trigger an anti-tumor T-cell response [135].

Dendritic cell vaccines ICT-107 [144] and Audencel [145] were evaluated in phase 2 clinical trials, though neither demonstrated a significant OS benefit [145]. Autologous dendritic cell vaccine DCVax-L in combination with standard therapy was studied in a phase 3 trial that allowed patients in the placebo group to receive DCVax-L upon progression [146]. Initial PFS analysis demonstrated no benefit from vaccine [146]. In a subsequent analysis, the primary endpoint was changed to OS and study participants were compared to external controls [147]. Though a modest OS benefit was reported compared to external controls (19.3 vs. 16.5 months), this study was limited by the inclusion of external controls from a variety of clinical trials that used different inclusion criteria and post-hoc change of the primary endpoint during secondary analysis [148].

#### Individualized Vaccines

Personalized neoantigen vaccines are derived from genetic sequencing of individual tumors [125]. Examples of such vaccines studied for GBM include APVAC1 (derived from known tumor antigens) and APVAC2 (personalized based on neoepitopes from individual tumors), studied in a phase 1 trial with promising preliminary OS results (NCT02149225) [149]. Another personalized vaccine NeoVax is under investigation in a phase 1 study for nGBM (NCT02287428) [150]. Personalized vaccines do present logistical challenges in terms of adequacy of tissue for vaccine development and time needed for preparation that will need to be addressed before widespread implementation can occur [134].

Adjuvant therapies to enhance efficacy of vaccines is under investigation. These strategies include combination of vaccine with ICIs, bevacizumab, or immunogenic cytokines such as interleukin-12 [134]. Optimizing delivery method may also enhance vaccine efficacy. As an example, Montanide, a water-in-oil emulsion, can prolong the release of vaccine antigens and has been shown to increase vaccine-related immune responses in 93% patients [134,151,152].

#### 4.2.3. Viral Oncolytics

Oncolytic viruses are natural or genetically engineered viruses that selectively infect, lyse, or replicate in tumor cells, generating a direct cytotoxic effect and triggering an anti-tumor immune response [109,153].

#### Adenoviral Vectors

Adenovirus has been utilized as a vector for oncolytic virotherapy for GBM. In a phase 2 trial of nGBM and other HGG, adenovirus vector AdV-tk was administered into the resection bed followed by valacyclovir administration with promising preliminary OS results (NCT00589875) [125]. The combination of AdV-tk/valacyclovir with nivolumab is under investigation (NCT03576612).

Other adenovirus vectors undergoing study for treatment of GBM include Ad-RTS-hIL-12 (NCT02026271), NSC-CRAd-S-pk7 (NCT03072134), and DNX-2401 (NCT00805376, NCT02798406) [154,155,156]. Combination of DNX-2401 with pembrolizumab is under investigation (NCT00805376) [157].

#### Herpes Simplex Virus (HSV) Vectors

HSV G47Δ is an oncolytic HSV1 that replicates in cancer cells, including GBM stem cells [158]. In a phase 2 trial for residual or rGBM, intratumoral injection was associated with an OS of 28.8 months from initial resection and 20.2 months from G47Δ initiation [153]. Another HSV1 oncolytic virus, rQNestin34.5v.2., is being investigated for the use of cyclophosphamide followed by intratumoral rQNestin34.5v.2 (NCT03152318).

#### Polio Viral Vectors

A recombinant polio virus, PVSRIPO, has been studied in GBM [125,139]. In a phase 1 study (NCT01491893), PVSRIPO was associated with favorable OS rates compared to historical controls [159]. LUMINOS-101 is a phase 2, multicenter single-arm study of intratumoral PVSRIPO followed by pembrolizumab in adults with rGBM (NCT04479241) with results pending.

#### Retroviral Vector

Vocimagene amiretrorepvec (Toca511) is a retrovirus that delivers complementary DNA (cDNA) for cytosine deaminase, rendering cells chemosensitive to 5-fluorocytosine. In a phase 3 trial, Toca511 conferred no OS benefit compared to standard therapy (11.1 vs. 12.2 months, NCT02414165) [126,160].

#### Measles Viral Vector

Administration of a carcinoembryonic antigen-expressing oncolytic measles virus (MV-CEA) in 22 patients with recurrent grade III or IV glioma was found to be well-tolerated [144]. MV-CEA plus anti-PD1 ICI therapy has been studied in mouse models with promising results [161,162,163].

To date, most clinical trials of oncolytic viral therapies are phase 1 studies assessing toxicity and tolerability and the only large-scale oncolytic virotherapy trial in GBM with Toca511 was negative. Optimal delivery mechanism, dosing, and timing remains to be explored and the degree of uptake of viral oncolytic therapies by tumors is uncertain. Triggering a sustainable antitumor immune response against the “immunologically cold” microenvironment of GBM remains problematic [153]. Combinatorial strategies are likely to be needed to achieve durable responses from virotherapy in GBM [153].

#### 4.2.4. CAR T-Cell Therapy

CAR T-cells are engineered to express an extracellular, tumor antigen-recognition domain coupled with an intracellular signaling domain that activates the T-cell upon antigen binding, triggering a tumor directed, T-cell mediated immune response [125,164]. This approach has been highly effective in hematologic malignancies and remains under investigation for GBM, so far with limited successes [165,166]. In a first in-human study of CART-EGFRvIII cells administered intravenously to 10 patients with rGBM, median OS was 8 months [167]. The study raises the question of whether targeting EGFRvIII alone will lead to a durable benefit, since antigen escape may occur frequently and rapidly [167].

Another phase 1 clinical trial administered CAR-EFGRvIII after lymphodepleting chemotherapy with intravenous interleukin-2 (IL-2) to 18 patients with recurrent GBM expressing EGFRvIII. One patient experienced treatment-related mortality after CAR T-cell administration at the highest dose: median PFS was 1.3 months; median OS was 6.9 months [168]. An additional EGFRvIII targeting CAR T trial with CARv3-TEAM-E via Ommaya reservoir was associated with dramatic radiographic responses in all three treated patients, but the effect was transient in two patients [169,170].

Another CAR T-cell target investigated for GBM is the interleukin-13 receptor α2 (IL13Rα2) [125]. In a pilot study of three patients with CAR T-cells targeting IL13Rα2 via intracranial catheter yielded transient responses in two patients [171]. A patient with recurrent multifocal leptomeningeal GBM was treated intraventricularly with multiple infusions of a second generation of CAR T-cells targeting IL13Rα2, which resulted in dramatic radiographic response, supporting further exploration of IL13Rα2 targeted CAR T-cell therapy [172]. In another phase 1 clinical trial (NCT05168423), six patients with rGBM were treated with intrathecal CAR T-cells, targeting both IL13Rα2 and EGFR to address the challenge of tumor heterogeneity [42]. All six patients had reduction in their tumor enhancement and size on interval brain MRI obtained 24–48 h after administration, although none met criteria for an objective radiographic response. Three of the four patients who had at least 2 months of follow up had SD. Other CAR T-cell therapies under investigation for GBM have targeted HER2 and erythropoietin-producing human hepatocellular carcinoma receptor (Eph). In a study of HER2 directed CAR T-cell therapy, of 16 evaluable patients with GBM, eight had clinical benefit (1 PR, 7 SD) [173]. In a pilot trial of [174,175] EphA2-CAR T-cells (NCT 03423992) intravenously administered a single dose, one patient had SD and two had progressive disease [175].

Challenges of CAR T-cell therapy in GBM include increased target cell engagement, antigen escape, and immunosuppressive microenvironment associated with GBM [125]. A viable strategy may be to use CAR T-cells targeting HER2, IL13Ra2, and EphA2, which significantly prolonged survival in mouse models [176]. Other challenges include low proliferation of CAR T-cells, which could be overcome by manufacturing cells that secrete proinflammatory cytokines, but this must be balanced with excessive inflammation that may have negative neurologic and systemic consequences [125,177].

#### 4.2.5. Cytokine Therapy

Standard of care therapy for GBM is associated with severe, prolonged systemic lymphopenia in about 40% of patients, correlating with decreased patient survival. Cytokine therapy may mitigate this lymphopenia and prolong survival. As an example, interleukin-7 (IL-7) stimulates T-cell homeostasis and proliferation [178,179,180]. NT-I7, a long-acting form of IL-7 demonstrated increased absolute lymphocyte counts [178]. A phase 2 randomized clinical trial is ongoing to evaluate the efficacy of NT-I7 vs. placebo on survival in patients with nGBM (NCT03687957).

### 4.3. Therapy Targeting DNA Repair Pathways

Cancer cells can resist DNA damage from radiation or alkylator therapy through activation of DNA damage checkpoint responses [181]. Therefore, targeting cancer cells’ ability to repair DNA represents a potential novel treatment strategy.

There are multiple mechanisms of DNA damage response (DDR) in tumor cells. In GBM, the most extensively studied group of agents that inhibit DDRs is poly (ADP-ribose) polymerase (PARP) inhibitors, which halt single strand DNA repair [182]. Veliparib, a PARP inhibitor, was studied in MGMT-methylated nGBM in combination with TMZ but did not confer an OS benefit (NCT02152982). Other PARP inhibitors, including pamiparib and olaparib, are undergoing study in nGBM and rGBM (NCT04614909).

Ataxia-telangiectasia mutated (ATM) is currently under evaluation as a DDR target via AZD1390 for radiosensitization (NCT03423628) [183]. DNA-PK (NTC02977780) and WEE1 (NCT01849146) are other novel DDR modifiers undergoing clinical trials [184].

### 4.4. Repurposed Medications

Medications primarily utilized for treatment of conditions other than malignancy have been considered for their potential anti-neoplastic effects for a variety of cancers including GBM. These repurposed medications are intended to exploit novel mechanisms to arrest tumor growth.

#### 4.4.1. Metformin

Metformin, a biguanide drug utilized to decrease gluconeogenesis in type 2 diabetes mellitus, may influence the pathogenesis of various tumors [185]. In preclinical studies, metformin in combination with 3-hydroxy-3-methyl-glutaryl-coenzyme A reductase inhibitor atorvastatin showed anti-tumor activity in GBM cell lines [186]. In a subsequent clinical trial (KNOG-1501), 81 patients were randomized to TMZ with metformin or placebo [187]. Although the TMZ and metformin therapy was well tolerated, there was no OS benefit [187].

#### 4.4.2. Fluoxetine

Fluoxetine is a selective serotonin reuptake inhibitor (SSRI) used for treatment of anxiety and depression. Fluoxetine was studied in GBM due to its ability to bind to α-amino-3-hydroxy-5-methyl-4-isoxazolepropionic acid receptors (AMPAR), a receptor exclusively expressed in glioma cells [188]. AMPAR binding induces an influx of intracellular calcium, which ultimately triggers cellular apoptosis [188]. In vivo studies have suggested fluoxetine may inhibit the growth of GBM in the brains of mice [188]. Retrospective analyses have additionally identified that GBM patients who were subsequently on fluoxetine as opposed to other SSRIs had improved survival [189]. However, the potential benefit of fluoxetine for treatment of GBM has not been evaluated in a prospective clinical trial.

#### 4.4.3. Gabapentin

Gabapentin, a frequently prescribed neuropathic pain medication, is of interest in patients with GBM as gabapentin inhibits the synaptogenic protein thrombospondin-1 (TSP-1), a protein responsible for GBM integration with local neural circuitry [190]. In a retrospective study evaluating GBM patients who had been treated with gabapentin in addition to TMZ, median OS was 20.8 months for gabapentin-treated patients compared to 14.7 months for patients who did not receive the agent (*p* = 0.002) [190]. However, these findings need prospective validation.

#### 4.4.4. CBD/THC

Cannabis and its various derivatives have been studied in cancer patients, including patients with GBM. Cannabinoid receptors CB1 and CB2 are expressed in gliomas [191,192]. In TMZ-resistant cell lines, when TMZ was used in combination with tetrahydrocannabinol (THC) there was a reduction in tumor cell proliferation [193]. In a subsequent study, intracranial THC administration in the surgical cavity of GBM patients led to a decrease in tumor cell proliferation in two of nine patients, but this was not associated with an OS benefit [194]. In a phase 1b study, nabiximols were well tolerated [195]. While an OS improvement was suggested for patients receiving nabiximols, the study was not adequately powered to detect a survival difference and these results should be interpreted with caution [195]. At this time, the role of cannabinoids in patients with GBM remains unclear.

#### 4.4.5. Glutamate Signaling Inhibitors

As GBM cells secrete copious quantities of glutamate, which drives epilepsy and tumor growth, inhibiting glutamate signaling is an active area of research in GBM. Various medications that play a role in glutamate signaling are undergoing study in GLUGLIO, a 1:1 randomized Ib/II multicenter trial of gabapentin, sulfasalazine, memantine, and chemoradiotherapy (Arm A) versus chemotherapy alone (Arm B) in patients with nGBM with a primary endpoint evaluating PFS-6 [196]. No results are available yet from this study.

## 5. Innovations in Trial Design

Given the limited efficacy of standard therapies, every patient with nGBM or rGBM should be considered for participation in a clinical trial. Despite this, accrual of patients into clinical trials is low with only 11% of GBM patients participating in trials [121,126]. Barriers to participation include need for travel to permit participation, lack of awareness of available studies, and overly stringent eligibility rules [121].

One approach to improving clinical trial accrual is using telehealth technology. In the United States, 80% of oncology patients are treated in a local community practice rather than an academic center [197]. Decentralized clinical trials centered around the patient rather than a physical institution is a novel approach to increase patient participation in clinical trials. For example, an open-label decentralized clinical trial that utilizes medication delivery by mail and remote phlebotomy services is assessing the use of a computer-based cognitive testing platform, wearable device technology for health promotion, and the effect of metformin on radiation-related toxicity in patients with CNS malignancies (NCT06377696). Results of this study may guide future decentralized clinical trial design for other clinical questions in neuro-oncology and modern medicine in general.

Innovations in clinical trial design have also allowed for more efficient testing of multiple drugs and combinations of drugs. While traditional clinical trials compare the efficacy of a single drug to the standard of care, adaptive trials save time and funds through application of data to the next patient in real time [154]. The Individualized Screening Trial of Innovative Glioblastoma Therapy (INSIGHhT) is a phase II platform trial that uses adaptive randomization and gene profiling to identify novel therapies for phase III testing in patients with newly diagnosed glioblastoma, MGMT unmethylated. Three experimental arms for abemaciclib, neratinib, and CC-115 were all compared simultaneously against a shared control arm. While patients were initially randomly assigned 1:1:1:1 between control and experimental arms, subsequent Bayesian adaptive randomization was incorporated based on biomarker-related progression-free survival [198]. Although the trial showed insufficient OS benefit to warrant phase III testing for the three drugs, the design answered this question with one trial instead of three separate phase II trials while maximizing the chance that patients would be exposed to a promising experimental therapy (NCT02977780).

GBM AGILE is another multiarm platform phase 2/3 trial that utilizes a shared control arm and Bayesian adaptive randomization for patients with nGBM and rGBM. The trial investigates multiple new therapies with the goal of matching therapies with patient biomarker subtypes. Patients are assigned via Bayesian adaptive randomization to arms based on their performance (NCT03970447).

The Neuro Master Match (N2M2) trial is another open-label, multicenter trial for nGBM that employs an umbrella design with five sub-trials, studying the effect of alectinib, idasanutlin, palbociclib, vismodegib, and temsirolimus according to tumor molecular profiles. Patients without matching alterations are randomized between sub-trials without strong biomarkers using atezolizumab and asunercept as standard of care. These novel approaches may efficiently identify subgroups of patients that respond to novel therapies [199].

Efforts to further improve the quality and reliability of clinical research include improvements in tumor response assessment with the goal of correctly identifying radiographic responses and OS benefits while minimizing confounders such as pseudoresponse, which can be seen in the setting of antiangiogenic use, and pseudoprogression, that can occur with immunotherapy. To that end, RANO criteria were developed and subsequently iteratively updated [200]. More recently, RANO 2.0 criteria were developed, introducing changes that will help ensure consistency between trials and facilitate comparisons [200]. As an example, use of post-radiotherapy MRI rather than post-operative MRI was recommended for standardization of comparisons [200]. Due to the frequent occurrence of treatment-related pseudoprogression during the 12 weeks immediately following radiation therapy, confirmatory imaging or histopathologic evidence of progression was suggested if adverse imaging changes are noted during this period within the treatment field [200]. In summary, limited participation of patients with GBM in trials slows completion of studies and potentially delays arrival of urgently needed novel therapeutics to the clinic. Innovative clinical trial designs that decentralize the process to reach more patients and reduce administrative burden may allow for high enrollment and real-time collection of an abundance of data that will advance the treatment of GBM. Clinical trials are evolving for the better, exposing more patients to potentially helpful therapeutics, using a single control group to compare against multiple combinations and therapies, operating under more standardized exclusion and response assessment criteria, and utilizing adaptive randomization and patient biomarker data to increase the likelihood of a successful match between patient and treatment.

## 6. Conclusions

Given the dismal prognosis of GBM, extensive research efforts remain underway in all areas of GBM care. This includes a wide variety of surgical innovations, radiotherapy innovations, systemic therapy innovations, and clinical trial design innovations utilizing anatomic, genetic, and immunologic advancements to create novel therapies, which are summarized in Table 4. Researchers remain motivated to improve patient outcomes and encourage patients to enroll in clinical trials to further therapies available for GBM.

## Figures and Tables

**Table 1 ijms-25-10570-t001:** Description of varying surgical innovations utilized in glioblastoma.

Surgical Innovation	Description/Utility
Intraoperative MRI	Defines location of anatomical structures prior to and during surgery. Ensures optimal resection prior to closure.
Fluorescence-Guided Surgery	Differentiation of tumor from brain. Improves EOR.
5-ALA	Visualization of non-enhancing and enhancing tumor.
Sodium fluorescein	Visualization of enhancing tumor.
Intraoperative Mapping	Minimizes post-operative neurologic deficits by identifying function including speech, cognitive, and motor processes via awake craniotomy and/or electrical stimulation.
Confocal Microscopy	Identifies glioma tissue in real time.
Phase 0/Window-of-Opportunity Studies	Evaluates intratumoral drug concentrations and pharmacodynamic impact of novel agents administered prior to surgery, potentially providing preliminary information on in vivo effectiveness.
Longitudinal CSF Access via CSF access devices or LPs	Longitudinal access to the tumor/tumor microenvironment for disease monitoring and evaluation of the biological impact of therapies
Local Drug Delivery	Reduces systemic toxicity, targeted distribution, and bypass of BBB and blood–ependymal barrier.
Focused Ultrasound	Used to kill tumor cells via sonodynamic therapy or to disrupt the BBB (increases CNS penetration of chemotherapy or increases diffusion of tumor biomarkers into blood stream for monitoring).
LITT	Damages cancer cells through heat; minimally invasive. Enhances BBB permeability.

EOR: extent of resection; 5-ALA: 5-aminolevulinic acid; LPs: lumbar punctures; CNS: central nervous system; LITT: laser interstitial thermal therapy.

**Table 3 ijms-25-10570-t003:** Review of systemic therapy clinical trials in glioblastoma.

Systemic Therapy Category	Agents	Selected Ongoing/Completed Clinical Trials	Status
Molecularly Targeted Therapies			
BRAF	Dabrafenib/trametinib	NCT02034110	Completed, results available
	Binimetinib/encorafenib	NCT01909453	Ongoing, results available
NTRK	Entrectinib	NCT02568267	Ongoing, recruiting complete
	Loretrectinib	NCT02465060	Ongoing, recruiting complete
	Loretrectinib	NCT04142437	Ongoing, actively recruiting
EGFR	ERAS-801	NCT05222802	Ongoing, recruiting complete
	WSD0922-Fu	NCT04197934	Ongoing, actively recruiting
Immunotherapies			
ICI	Nivolumab versus bevacizumab	NCT02017717	Completed, results available
	RT + TMZ + nivolumab or placebo	NCT02667587	Completed, results available
	RT+ nivolumab vs. RT+ TMZ	NCT02617589	Completed, results available
	Pembrolizumab vs. Pembrolizumab + bevacizumab	NCT02337491	Completed, results available
	Pembrolizumab in hypermutated recurrent malignant gliomas	NCT02658279	Ongoing, not recruiting
	Ipilimumab + nivolumab in recurrent gliomas with elevated mutational burden	NCT04145115	Ongoing, recruitment suspended
	LITT + pembrolizumab	NCT03277638	Ongoing, actively recruiting
Vaccines	Rindopepimut (EGFRvIII-targeting peptide vaccine) + TMZ	NCT01480479	Completed, results available
	SurVaxM (survivin-targeting peptide vaccine) + pembrolizumab	NCT04013672	Ongoing, not recruiting
	VXM01 + avelumab	NCT03750071	Ongoing, not recruiting
	IMA950/polyICLC	NCT01920191	Completed, results available
	IMA950/poly-ICLC with or without pembrolizumab	NCT03665545	Ongoing, not recruiting
	EO2401	NCT04116658	Ongoing, not recruiting
	Pp65 vaccine + TMZ	NCT00639639	Completed, results available
	ICT-107 dendritic cell vaccine	NCT01280552	Completed, results available
	Personalized neoantigen vaccines (APVAC1, APVAC2)	NCT02149225	Completed, results available
Oncolytic virotherapy	AdV-tk into tumor resection bed during surgery	NCT00589875	Completed, results available
	AdV-tk + valacyclovir + nivolumab	NCT03576612	Ongoing, not recruiting
	Ad-RTS-hIL-12 intratumorally injected	NCT02026271	Completed, results available
	NSC-CRAd-S-pk7 injected into walls of resection cavity	NCT03072134	Completed, results available
	DNX-2401	NCT00805376	Completed, results available
	DNX-2401+ pembrolizumab	NCT02798406	Completed, results available
	HSV G47Δ	UMIN000002661	Completed, results available
	G47Δ injected intratumorally in patients with residual or recurrent GBM	UMIN000015995	Completed, results available
	Cyclophosphamide followed by intratumoral rQNestin34.5v.2	NCT03152318	Ongoing, actively recruiting
	PVSRIPO	NCT01491893	Completed, results available
	PVSRIPO followed by pembrolizumab	NCT04479241	Ongoing, not recruiting
	Toca511 phase I	NCT01470794	Completed, results available
	Toca511 phase III	NCT02414165	Completed, results available
	MV-CEA	NCT00390299	Completed, results available
CAR T-Cell therapy	CARv3-TEAM-E via Ommaya reservoir	NCT05660369	Completed, results available
	Intrathecal CAR T-cells, targeting both IL13Rα2 and EGFR	NCT05168423	Completed, results available
	EphA2-CAR T-cells intravenously with prior fludarabine and cyclophosphamide	NCT03423992	Completed, results available
Cytokine therapy	NT-17 vs. placebo, phase II	NCT03687957	Ongoing, actively recruiting
	NT-17 after chemoradiation, phase I/II	NCT03687957	Completed, results available
DNA Repair Therapies			
PARP	Olaparib	NCT04614909	Ongoing, actively recruiting
ATM	AZD1390	NCT03423628	Ongoing, actively recruiting
DNA-PK	CC115	NCT02977780	Ongoing, actively recruiting
WEE-1	Adavosertib	NCT01849146	Ongoing, not recruiting

**Table 4 ijms-25-10570-t004:** Review of ongoing novel GBM therapies.

Innovations in Surgery	Innovations in Radiation Therapy	Innovations in Systemic Therapy
Improved tumor visualization 5-ALA Sodium fluorescein	Particle based therapies Proton Carbon Ion BNCT	Targeted Therapy Targeted molecular therapies (BRAF, NTRK, EGFR, PI3K-mTOR, VEGF) DNA Repair Pathways (PARP, ATM)
Techniques to improve maximal safe surgical resection Intraoperative MRI Confocal microscopy Raman histology	Local Therapies IORT IBT SRS	Immunotherapy ICI Vaccine-based therapy Viral oncolytics CAR T-Cell Therapy Cytokine therapy
Improvements in BBB penetration and tumoral access Phase 0 studies Longitudinal CSF access Local drug delivery FUS LITT		Repurposed Medications Metformin Fluoxetine Gabapentin THC Glutamate signaling inhibitors

5-ALA: 5-ami-nolevulinic acid; FUS: focused ultrasound therapy; LITT: laser interstitial therapy; BNCT: boron neuron capture therapy; IORT: intraoperative radiotherapy; IBT: interstitial brachytherapy; SRS: stereotactic radiosurgery; BRAF: V-RAF murine viral oncogene homolog B1; NTRK: neurotrophic tyrosine kinase inhibitor; EGFR: epidermal growth factor receptor; PI3K-mTOR: phosphatase and tensin homolog; VEGF: vascular endothelial growth factor; PARP: poly-ADP-ribose polymerase; ATM: ataxia-telangiectasia mutated; ICI: immune checkpoint inhibitors; CAR: chimeric antigen receptor.

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
