# Peer review of "Review of Novel Surgical, Radiation, and Systemic Therapies and Clinical Trials in Glioblastoma"

_ijms, 2024, doi:10.3390/ijms251910570_

Round 1

Reviewer 1 Report

Comments and Suggestions for Authors

I was pleased to be asked to review this paper as there is certainly a need for a paper such as this. As the authors state, glioblastoma prognosis is poor and there are many innovations and developments in the management which this paper reviews. The innovations in trial design are interesting and well described. The authors have generated a useful and comprehensive review. It would benefit from some reorganisation and expansion for the International Journal. A few points are as below:

1. In describing novel treatments it would be helpful to expand on the standard treatments.These standard treatments do vary across the world both for primary presentation of glioblastoma and recurrence. For example Tumour treating Fields (TTF) is not universally standard and may be considered as a novel therapy and so would benefit from further explanation and review.

2. It would be helpful to discuss the management of recurrence of glioblastoma, again as a baseline for many of the novel therapies discussed which are usually initially used in patients with recurrence. eg PCV/lomustine. Also the use of reirradiation in recurrence as discussed in 3.3

3. The Brem study for use of carmustine wafers -gliadel in recurrence - did show a significant survival benefit. (paragraph 1 introduction )

4. The use of repurposed drugs as well as metformin, which is mentioned,  could be expanded further.

Otherwise I would recommend publication after a few minor revisions as above.

Author Response

  1. In describing novel treatments it would be helpful to expand on the standard treatments. These standard treatments do vary across the world both for primary presentation of glioblastoma and recurrence. For example, Tumour treating Fields (TTF) is not universally standard and may be considered as a novel therapy and so would benefit from further explanation and review.
    1. GBM and its standard therapy was expanded in the introduction paragraph. Discussion regarding TTF was also further expanded in the introduction paragraph as although TTF are not available everywhere, they are FDA approved for initial treatment in newly diagnosed GBM. Recurrence management is also now introduced in the introduction paragraph.
  2. It would be helpful to discuss the management of recurrence of glioblastoma, again as a baseline for many of the novel therapies discussed which are usually initially used in patients with recurrence. eg PCV/lomustine. Also the use of reirradiation in recurrence as discussed in 3.3
    1. This was added in the introduction paragraph.
  3. The Brem study for use of carmustine wafers -gliadel in recurrence - did show a significant survival benefit. (paragraph 1 introduction )
    1. Thank you for pointing out this research. This has been added in the introductory paragraph. 
  4. The use of repurposed drugs as well as metformin, which is mentioned,  could be expanded further.
    1. Thank you for this suggestion. A section titled “repurposed medications” was added under the novel systemic therapy sections.

Reviewer 2 Report

Comments and Suggestions for Authors

The authors here provide a systematic summary of the innovative GBM therapies. These new therapies are getting more attention in the clinical tirals with predictable expectations. Overall, this review is well organized by the innovations in Surgical Management, Radiation Therapy, Systemic Therapy and Clinical trial design.

In contrast with the GBM therapies, the introduction of GBM diseases, pathogenesis and current therapies is not enough discussed. The novelty of GBM therapies is not well shown in each part, which should be further emphasized.

Also, there is less illustrative figures to show the principle of novel therapies, at least for some outstanding GBM therapies. The pros and cons of these novel therapies should be also shown in a separate paragraph.

“5-ALA” and “Sodium Fluorescein” in the Table 1 is shown bold and left in inconsistent place. Why?

Author Response

  1. In contrast with the GBM therapies, the introduction of GBM diseases, pathogenesis and current therapies is not enough discussed. The novelty of GBM therapies is not well shown in each part, which should be further emphasized. 
    1. The introduction has been expanded to better introduce GBM and standard therapies. We additionally have expanded the discussion throughout the paper to better emphasize the novelty of the discussed therapies and how their mechanisms differ from standard therapy (i.e. see surgical management introduction, targeted molecular therapies, immunotherapy introduction).
  2. Also, there is less illustrative figures to show the principle of novel therapies, at least for some outstanding GBM therapies. The pros and cons of these novel therapies should be also shown in a separate paragraph. 
    1. Table 4 was added to provide a simplified overview and review the discussed novel therapies.
    2. We have woven pros and cons throughout the paper in the body of the paragraph for each specific therapy. As there are a very large variety of novel therapies discussed, a separate paragraph regarding pros and cons per therapy would unfortunately significantly lengthen an already long paper.
  3. “5-ALA” and “Sodium Fluorescein” in the Table 1 is shown bold and left in inconsistent place. Why?
    1. These are intended to be a sub headers of “fluorescence guided therapy”. The formatting has been adjusted to make this clear.

Round 2

Reviewer 2 Report

Comments and Suggestions for Authors

This revised manuscript has been much improved for the publication in IJMS.